# LncRNA/miRNA/mRNA Network Introduces Novel Biomarkers in Prostate Cancer

**DOI:** 10.3390/cells11233776

**Published:** 2022-11-25

**Authors:** Mohammad Taheri, Arash Safarzadeh, Bashdar Mahmud Hussen, Soudeh Ghafouri-Fard, Aria Baniahmad

**Affiliations:** 1Institute of Human Genetics, Jena University Hospital, 07743 Jena, Germany; 2Urology and Nephrology Research Center, Shahid Beheshti University of Medical Sciences, Tehran 16666-63111, Iran; 3Department of Medical Genetics, School of Medicine, Shahid Beheshti University of Medical Sciences, Tehran 19835-35511, Iran; 4Department of Biomedical Sciences, Cihan University-Erbil, Kurdistan Region, Erbil 44001, Iraq; 5Department of Pharmacognosy, College of Pharmacy, Hawler Medical University, Kurdistan Region, Erbil 44001, Iraq

**Keywords:** prostate cancer, ceRNA, lncRNA, miRNA, regulatory network

## Abstract

The construction of a competing endogenous RNA (ceRNA) network is an important step in the identification of the role of differentially expressed genes in cancers. In the current research, we used a number of bioinformatics tools to construct the ceRNA network in prostate cancer and identify the importance of these modules in predicting the survival of patients with this type of cancer. An assessment of microarray data of prostate cancer and normal samples using the Limma package led to the identification of differential expressed (DE) RNAs that we stratified into mRNA, lncRNA, and miRNAs, resulting in 684 DEmRNAs, including 437 downregulated DEmRNAs (such as TGM4 and SCGB1A1) and 241 upregulated DEmRNAs (such as TDRD1 and CRISP3); 6 DElncRNAs, including 1 downregulated DElncRNA (H19) and 5 upregulated DElncRNAs (such as PCA3 and PCGEM1); and 59 DEmiRNAs, including 30 downregulated DEmiRNAs (such as hsa-miR-1274a and hsa-miR-1274b) and 29 upregulated DEmiRNAs (such as hsa-miR-1268 and hsa-miR-1207-5p). The ceRNA network contained a total of 5 miRNAs, 5 lncRNAs, and 17 mRNAs. We identified hsa-miR-17, hsa-miR-93, hsa-miR-150, hsa-miR-25, PART1, hsa-miR-125b, PCA3, H19, RND3, and ITGB8 as the 10 hub genes in the ceRNA network. According to the ROC analysis, the expression levels of 19 hub genes showed a high diagnostic value. Taken together, we introduce a number of novel promising diagnostic biomarkers for prostate cancer.

## 1. Introduction

Prostate cancer is the most frequent malignancy in males worldwide, accounting for 27% of all cancer diagnoses [1]. During the period from 2014 to 2018, the incidence of prostate cancer has stayed stable, yet the annual incidence for advanced prostate cancer has increased by 4–6% since 2011 [1]. This means that the percentage of prostate cancer cases being diagnosed at advanced stages has increased by approximately two times over the past decade. Consequently, it is necessary to find appropriate markers for an early diagnosis of this common cancer. The prostate-specific antigen (PSA), as the mostly used marker, has some limitations, including the poor interchangeability of PSA results which are acquired from diverse tests. The current gaps should be filled by arranging commutable reference materials for calibration immunoassay tests, identifying analytical features that can clarify the different performance of assays, and giving more focus on laboratory tests when clinical practice guidelines are organized [2].

It is also important to mention that high-grade prostate cancer is likely on a genetic basis, whereas low-grade disease is more likely to be associated with environmental factors. In fact, the vivid alteration in stage from the time when PSA was introduced as a screening test has been accompanied by a more modest shift in Gleason grade, suggesting that grade may be established in the early phases of tumor pathogenesis [3]. Meanwhile, metastatic prostate cancers exhibit histological and immunophenotypical heterogeneity, necessitating the conduction of individualized therapeutic regimens [4]. For reasons of cost-effectiveness, this panel should be applied to individuals at risk of high-grade prostate cancer, likely based on PSA levels. According to current guideline recommendations, more recently, a PSA-based algorithm for predicting high-grade tumors has been developed [5].

The recent decade has witnessed a revolution in the application of high throughput data analysis tools, leading to the construction of several interaction networks between different types of biomolecules including RNA, DNA, and protein in different disease contexts, particularly cancer. This type of study has led to the identification of promising biomarkers for the early detection of cancer. In the field of prostate cancer, several efforts have been made. For instance, the re-analysis of high throughput expression data has led to the identification of several differentially expressed genes (DEGs) between cancerous and normal tissues. Further functional enrichment analyses have shown the relationship between these genes and clinical outcomes of patients [6]. Similar approaches have identified differentially expressed circular RNAs in this type of cancer and the main signaling pathways being controlled by these transcripts [7].

The competing endogenous RNA (ceRNA) network being constructed between long non-coding RNAs (lncRNAs), microRNAs (miRNAs), and mRNAs has been shown to contribute to the pathoetiology of different cancers and regulate fundamental processes, both within cancer cells and inside the tumor microenvironment [8,9]. Identification of this type of interaction between different RNA molecules can introduce novel biomarkers for cancer diagnosis. In the current research, we used a number of bioinformatics tools to construct the ceRNA network in prostate cancer and identify the importance of these modules in the prediction of survival of patients with this type of cancer.

## 2. Methods

### 2.1. Microarray Data Collection

The Gene Expression Omnibus (GEO; http://www.ncbi.nlm.nih.gov/geo/) (accessed on 12 September 2022) was used to obtain the human expression data for GSE88808 (Illumina HumanHT-12 WG-DASL V4.0 expression beadchip (gene summary, GenomeStudio report), Duarte, CA, USA), GSE200879 ((HTA-2_0) Affymetrix Human Transcriptome Array 2.0 (Genosplice), Créteil, France), and GSE60117 (Agilent-021827 Human miRNA Microarray (V3), Palo Alto, CA, USA). There were 98, 137, and 77 samples in each of these datasets, respectively. For further analysis, we chose 113 prostate cancers and 9 normal tissues from GSE200879, 49 prostate tumors and 49 adjacent normal samples from GSE88808, and 56 prostate tumors and 21 normal tissues from GSE60117 (Table 1). Moreover, we selected prostate tumor samples with Gleason scores of 7 or above. The expression data included the expression signatures of lncRNAs, miRNAs, and mRNAs.

### 2.2. Microarray Data Processing, Integrative Meta-Analysis and Assessment of Data Quality

Normalization is a crucial stage in the integration of heterogeneous data since the described datasets contain various and trendy platforms. The statistical programming language R was used for the processing and integration of microarray data. Data were initially normalized independently for pre-processing using the preprocessCore package’s normalizeQuantiles function (https://bioconductor.org/packages/release/bioc/html/preprocessCore.html (accessed on 14 September 2022). Next, data from both platforms were combined using the R software. The ComBat function from the R Package Surrogate Variable Analysis (SVA) was utilized to exclude batch effects (non-biological differences) [10]. The batch-effect removal was assessed using PCA and a boxplot. A unit expression matrix was the outcome of the meta-analysis (the combination of three datasets of this study).

### 2.3. Analysis of Differentially Expressed lncRNAs, miRNAs and mRNAs

We used the Limma package in R language [11] to identify differentially expressed mRNAs (DEmRNAs), lncRNAs (DElncRNAs), and miRNAs (DEmiRNAs) between prostate and normal samples. GSE88808 and GSE200879 were used to obtain DEmRNAs and DElncRNAs. GSE60117 was utilized to acquire DEmiRNAs. The cut-off criteria for evaluating DEGs were |log2 fold Change (FC)| > 0.5 and false discovery rate (FDR; adjusted *p* value) < 0.05. Following that, we used the HUGO gene nomenclature committee to identify DElncRNAs.

### 2.4. Two-Way Clustering of DEGs

We determined the gene expression levels of significant DEGs. We used this data in the pheatmap package in R language [12] to complete the two-way clustering based on the Euclidean distance.

### 2.5. TCGA Data Collection and Processing

We included a total of 500 PRAD samples and 52 control samples for further analysis. We used the TCGAbiolinks package to download the transcriptome profiling data (TCGA-PRAD), and limma and edgeR packages to analyze the data. As a result, DEGs were evaluated with the cut-off criteria of the false discovery rate (FDR; adjusted *p* value) < 0.05 and |log2 fold Change (FC)| > 0.5. Finally, we identified the genes that were present in both the TCGA and GEO datasets.

### 2.6. Gene Ontology (GO) Enrichment Analysis

We used the clusterProfiler R package [13] to perform gene ontology (GO) enrichment analysis to investigate the functions of the remarkably upregulated and downregulated DEGs that we discovered. The functional category criteria were established at an adjusted *p*-value of 0.05 or below.

### 2.7. Kyoto Encyclopedia of Genes and Genomes (KEGG) Pathway Analysis

To determine the potential roles of DEGs that took part in the pathways based on the KEGG database, KEGG pathway analyses of these genes were conducted [14].

### 2.8. PPI Network Construction and Hub Genes Identification

The PPI network for DEGs was built using the STRING database [15]. The highest level of confidence (confidence score > 0.9) and text mining, experiments, and database sources were used to establish the interactions parameter. The interactions between the proteins were examined using the Cytoscape software v3.9 [16]. Finally, the Cytohubba plugin [17] of Cytoscape was used to calculate the degree of connectivity of nodes to find the top 20 DEGs as hub genes. 

### 2.9. Regulatory Network of miRNA-Hub Genes and TF-Hub Genes

The Networkanalyst database [18] was used to create the relationships between the PPI hub genes and the transcription factors (TFs) and miRNAs. As a result, we identified TF and miRNA with the highest degree in the networks.

### 2.10. Constructing the ceRNA Network and Hub Genes Identification

We created a ceRNA network by carrying out the following steps: (1) Evaluating the interactions between lncRNAs and miRNAs based on the PC-related miRNAs using miRcode (http://www.mircode.org/) (accessed on 30 September 2022). (2) Applying miRDB (http://www.mirdb.org/) (accessed on 30 September 2022) [19], miRTarBase (https://mirtarbase.cuhk.edu.cn/) (accessed on 30 September 2022) [20], TargetScan (http://www.targetscan.org/) (accessed on 30 September 2022) [21] and miRWalk (http://129.206.7.150/) (accessed on 30 September 2022) [22] for the prediction of miRNA-targeted mRNAs. (3) Using Cytoscape v3.9, we identified the intersection of DE lncRNAs and mRNAs and created a lncRNA/mRNA/miRNA ceRNA network, and (4) utilized the degree method and the cytohubba plugin to predict hub genes.

### 2.11. Validation of Hub Genes via Expression Values and Receiver Operating Characteristic (ROC) Curve Analysis

The expression value of hub genes was assessed using the Gepia database [23]. The hub genes in the TCGA-PRAD RNA-seq data were examined, and those present in the PPI and ceRNA networks as well as in the TCGA-PRAD were chosen for gene expression validation. Additionally, the area under the curve (AUC) values derived from ROC curve analysis were used to assess the diagnostic efficacy of hub genes. Figure 1 demonstrates the workflow of the study.

## 3. Results

### 3.1. Microarray Data Processing, Integrative Meta-Analysis, and Assessment of Data Quality

The boxplot of unprocessed raw data and normalized data after batch effect removal is shown in Figure 2. These boxplots demonstrate the accuracy of the normalization and quality of the expression data. A total of 220 samples are displayed in the PCA plot (Figure 3) on the 2D plane covered by their first two main components (PC1 and PC2). This figure shows the integration of two samples and the elimination of the batch effect.

### 3.2. DEGs Analysis

The Limma package (version 3.52.3) was used to analyze microarray data from prostate cancer and normal samples. This analysis resulted in the identification of 684 DEmRNAs, including 437 downregulated DEmRNAs (such as TGM4 and SCGB1A1) and 241 upregulated DEmRNAs (such as TDRD1 and CRISP3); 6 DElncRNAs, including 1 downregulated DElncRNA (H19) and 5 upregulated DElncRNAs (such as PCA3 and PCGEM1); and 59 DEmiRNAs, including 30 downregulated DEmiRNAs (such as hsa-miR-1274a and hsa-miR-1274b) and 29 upregulated DEmiRNAs (such as hsa-miR-1268 and hsa-miR-1207-5p). Table 2, Table 3 and Table 4 present the most substantially up- and down-regulated DEmRNAs, DElncRNAs, and DEmiRNAs, respectively.

In order to compare the variation in miRNA, lncRNA, and mRNA expressions between prostate cancer and normal samples, we utilized the volcano plot using the EnhancedVolcano package [24] in R (Figure 4). In addition, two heatmaps demonstrated that 20 clearly distinct DEmRNA expression patterns between prostate and normal samples were identifiable (Figure 5A). The expression of DElncRNAs is also shown in a heatmap (Figure 5B).

### 3.3. TCGA Data Analysis

All available TCGA data on PRAD were obtained from the TCGA data portal using TCGAbiolinks package in R programming language. In September 2022, there were RNAseq data on 552 PRAD samples, including 500 primary solid tumor and 52 solid tissue normal samples. We analyzed this data using limma and edgeR packages to retrieve DEGs. DEGs were evaluated with the cut-off criteria of false discovery rate (FDR; adjusted *p* value) < 0.05 and |log2 fold Change (FC)| > 0.5. Finally, we identified the genes that exist in both GEO datasets and the TCGA dataset (Figure 6). As a result, we found out that there were 193 upregulated and 416 downregulated DEGs in both GEO datasets and the TCGA dataset. We continued the analysis with these genes.

### 3.4. GO Enrichment Analysis of DEGs

For the analysis, the clusterProfiler package (version 4.4.4) was employed. In GO functional enrichment analysis, 604 GO entries reached an adjusted *p* value of less than 0.05, most of which were biological processes (BP), followed by cellular component (CC) and molecular function (MF). The first 30 entries are collagen-containing extracellular matrix (CC), sarcolemma (CC), cell-substrate junction (CC), focal adhesion (CC), basement membrane (CC), extracellular matrix binding (MF), muscle system process (BP), muscle contraction (BP), muscle tissue development (BP), urogenital system development (BP), ear development (BP), membrane raft (CC), membrane microdomain (CC), gland development (BP), muscle cell differentiation (BP), contractile fiber (CC), regulation of epithelial cell proliferation (BP), cell-substrate adhesion (BP), epithelial cell proliferation (CC), response to xenobiotic stimulus (BP), mesenchyme development (BP), regulation of smooth muscle cell proliferation (BP), smooth muscle cell proliferation (BP), laminin binding (MF), cell junction assembly (BP), extracellular matrix structural constituent (MF), collagen binding (MF), inner ear development (BP), mesenchymal cell differentiation (BP), and muscle organ development (BP). Figure 7 shows the bar plots of the top 10 enriched functions.

The dot plot of the top 10 enriched functions and the enriched GO-induced graph are, respectively, visualized in Figure 8 and Figure 9.

Figure 10 indicates a network of GO terms and Figure 11 shows the gene-concept network of the top five GO terms.

In Figure 12, the intersection of the top 10 GO phrases was represented by the UpSet plot. It highlights the gene overlap between several gene sets.

### 3.5. Pathway Analysis

Using Pathview [25] and gage [26] packages in R, to find the probable functional genes, the KEGG pathway analysis of 241 upregulated and 437 downregulated DEGs was carried out (Table 5 and Figure 13).

### 3.6. Identification of Genes Related to Cell Senescence

We used the CellAge database (https://genomics.senescence.info/cells/ (accessed on 24 September 2022) to find the genes associated with cell senescence. A total of 279 genes associated with cell senescence were found in this database. As a result, 21 DEGs, including TP63, ID4, ITGB4, CAV1, ID1, TLR3, TGFB1I1, MYLK, NTN4, ETS2, LGALS3, GNG11, IGFBP3, SIK1, VEGFA, EZH2, MATK, HJURP, NOX4, MYC, and MMP9 were discovered to be involved in cell senescence (Figure 14).

### 3.7. PPI Network Construction and Selection of Hub Genes

To find the hub genes, a PPI network of DEGs (Figure 15) with 178 nodes and 185 edges created from STRING was imported into the Cytohubba plugin of Cytoscape 3.9. ITGA2, ITGA3, CAV1, PRKCA, ITGB4, ITGA8, MET, VEGFA, GPC1, MMP9, CAMK2B, LAMA3, PAK1, MYH11, and ITGB6 were the 15 hub genes with the highest degree of connectivity. Table 6 shows information about these hub genes. The greatest degree to the lowest degree is used to order these hubs.

### 3.8. Inspection of the Regulatory Network of miRNA-Hub Genes

The miRNAs that target hub genes were collected from the Networkanalyst web database (Figure 16). Both Hsa-miR-26b-5p and Hsa-miR-1-3p were considered important miRNAs because they interacted with hub genes at the highest level (degree 5) possible.

### 3.9. Examination of the Regulatory Network of TF-Hub Genes

By using the Networkanalyst database, we were able to acquire TFs that target hub genes (Figure 17). The TF-hub gene network revealed that SUZ12 regulates 12 hub genes and may play a significant role in the development of prostate cancer.

### 3.10. ceRNA Network Construction in Prostate Cancer

The interaction between lncRNAs and miRNAs was evaluated using miRcode. This step showed that 5 of the 6 lncRNAs may target 20 of the 59 PC-specific DEmiRNAs (Table 7). Then, to determine which mRNAs were targeted by these 20 miRNAs, we utilized miRWalk in combination with the miRTarBase, TargetScan, and miRDB filters. According to the research, 5 of the 684 mRNAs may be targeted by 5 miRNAs (Table 8). If miRNA-targeted mRNAs were not found in DEmRNAs, they were eliminated. The information from Table 7 and Table 8 was utilized to construct the lncRNA–miRNA–mRNA ceRNA network in Cytoscape 3.9. The ceRNA network contained a total of 5 miRNAs, 5 lncRNAs, and 17 mRNAs (Figure 18). We displayed this ceRNA network using a Sankey diagram generated by the ggalluvial R package (Version: 0.12.3) [27] in order to better understand the impact of lncRNAs on mRNAs in prostate, of which is mediated by their interaction with miRNAs (Figure 19). Finally, we determined the nodes’ degrees using the cytohubba app, and we displayed the top 10 nodes in the network with the highest degree centrality (Figure 20). As 10 hub genes in the ceRNA network, we identified hsa-miR-17, hsa-miR-93, hsa-miR-150, hsa-miR-25, PART1, hsa-miR-125b, PCA3, H19, RND3, and ITGB8.

### 3.11. Validation of Hub Genes via Expression Value

The expression value of hub genes was assessed using the Gepia (http://gepia.cancer-pku.cn/) (accessed on 30 September 2022). Additionally, we used CancerMIRNome (http://bioinfo.jialab-ucr.org/CancerMIRNome/) (accessed on 30 September 2022) to evaluate the hub miRNA gene expression value. As a result, CAMK2B, CAV1, GPC1, H19, ITGA3, ITGA8, ITGB4, LAMA3, MMP9, MYH11, PCA3, PRKCA, RND3, VEGFA, hsa-miR-25, hsa-miR-93, hsa-miR-125b, and hsa-miR-17 indicated good statistical significance (Figure 21).

### 3.12. Validation of Hub Genes via ROC Curve

We used graphpad prism 9.0 and CancerMIRNome to construct ROC curves. The ROC curve was used to evaluate how accurately the hub genes predicted outcomes. AUC was used to compare the diagnostic values of these hub genes. ROC curves and AUC values of the dataset are shown in Figure 22. The computed AUC values in this study, which were based on the findings, varied from 0.7 to 1—this is considered to have high discriminative power. The expression levels of 22 hub genes had a high diagnostic value, according to the ROC analysis.

## 4. Discussion

In order to find novel biomarkers for prostate cancer, we used a bioinformatics approach and constructed the ceRNA network in this context. We also valued the importance of the identified hub genes in the pathogenesis of prostate cancer and found their association with signaling pathways. 

In the first step, we identified several DEGS. TGM4 and SCGB1A1 have been among down-regulated mRNAs in this type of cancer. *TGM4* gene codes for transglutaminase 4, a protein with a restricted pattern of expression toward prostate. The encoded protein has been shown to regulate the interactions between prostate cancer cells and vascular endothelial cells through bypassing the ROCK pathway [28]. Moreover, TDRD1 and CRISP3 have been among upregulated mRNAs. *TDRD1* is responsible for coding a protein containing a tumor domain that suppresses transposable elements during spermatogenesis. The encoded protein has been shown to be expressed in most human prostate tumors, in spite of its absence in normal prostate tissues [29]. CRISP3 is found in low quantities in seminal plasma. The over-expression of CRISP3 in addition to the down-regulation of PTEN illustrates a subgroup of prostate cancer patients with a high probability of biochemical recurrence [30]. We also reported the down-regulation of H19 and up-regulation of a number of lncRNAs, such as PCA3 and PCGEM1. Notably, PCA3 is probably the most important lncRNA biomarker for prostate cancer [31].

Key factors known to be involved in prostate cancer tumorigenesis were identified with this approach. However, there are limitations. Importantly, the bioinformatic prediction is strongly dependent on the patient-derived data sets and number of data sets. We have used here a large number of data sets and from different original sources to minimize off-target finings. Still, it may be possible that using other patient-derived data sets will provide results that do not show a complete overlap of the ceRNA network.

KEGG pathway analysis has revealed glutathione metabolism and the regulation of actin cytoskeleton as the mostly down-regulated pathways. DEmiRNAs have also been reported to participate in a number of critical signaling pathways that affect prostate carcinogenesis. The ceRNA network contained a total of 5 miRNAs, 5 lncRNAs, and 17 mRNAs. We identified hsa-miR-17, hsa-miR-93, hsa-miR-150, hsa-miR-25, PART1, hsa-miR-125b, PCA3, H19, RND3, and ITGB8 as the 10 hub genes in the ceRNA network. According to the ROC analysis, the expression levels of 19 hub genes showed a high diagnostic value. Therefore, the constructed ceRNA network has been shown to affect important cellular pathways in prostate carcinogenesis and influence the prognosis of patients with this type of cancer. Taken together, we introduce a number of novel, promising diagnostic biomarkers for prostate cancer. This ceRNA-based panel can be applied as a second-level test to patients with certain PSA levels. This level should be identified in future studies.

## Figures and Tables

**Figure 1 cells-11-03776-f001:**
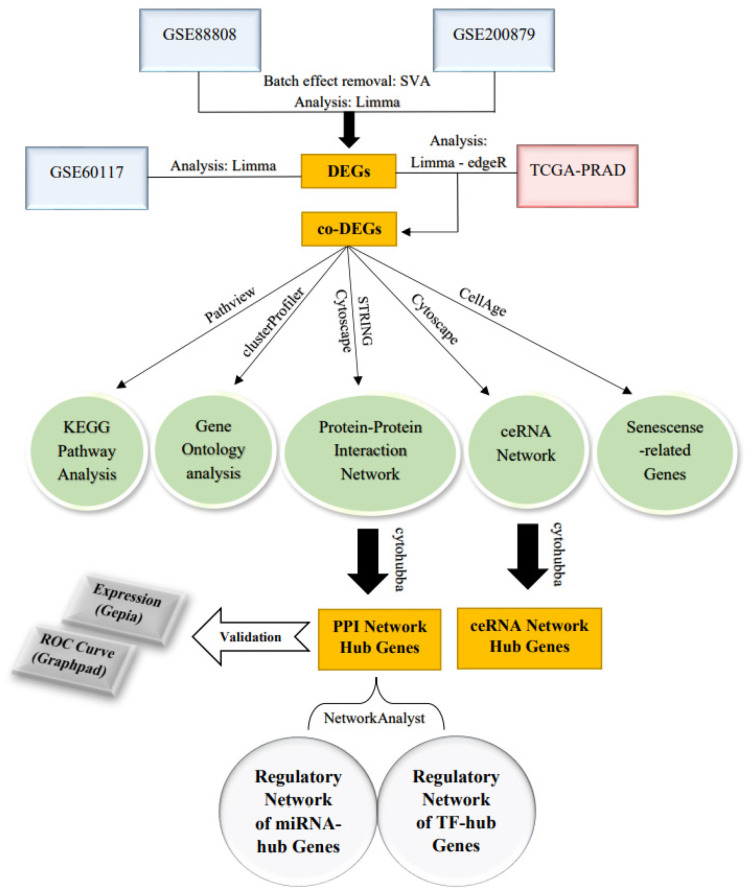
The workflow of the study.

**Figure 2 cells-11-03776-f002:**
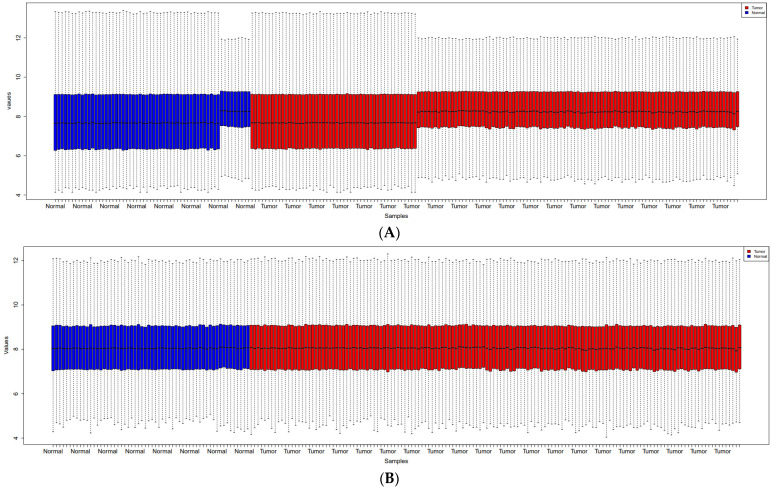
Boxplots show the raw data (**A**) and the normalized data following the removal of the batch effect (**B**). Prostate cancer samples are indicated by red boxes, whereas normal samples are shown by blue boxes.

**Figure 3 cells-11-03776-f003:**
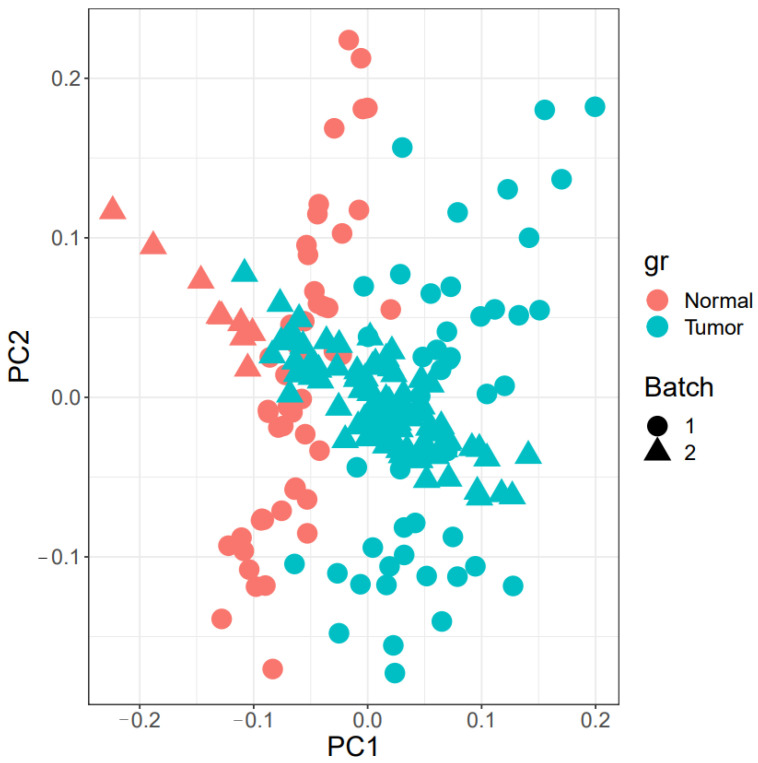
PCA plot. The batch implies that there are two platforms in the data. In addition, two groups of normal and tumor samples were created.

**Figure 4 cells-11-03776-f004:**
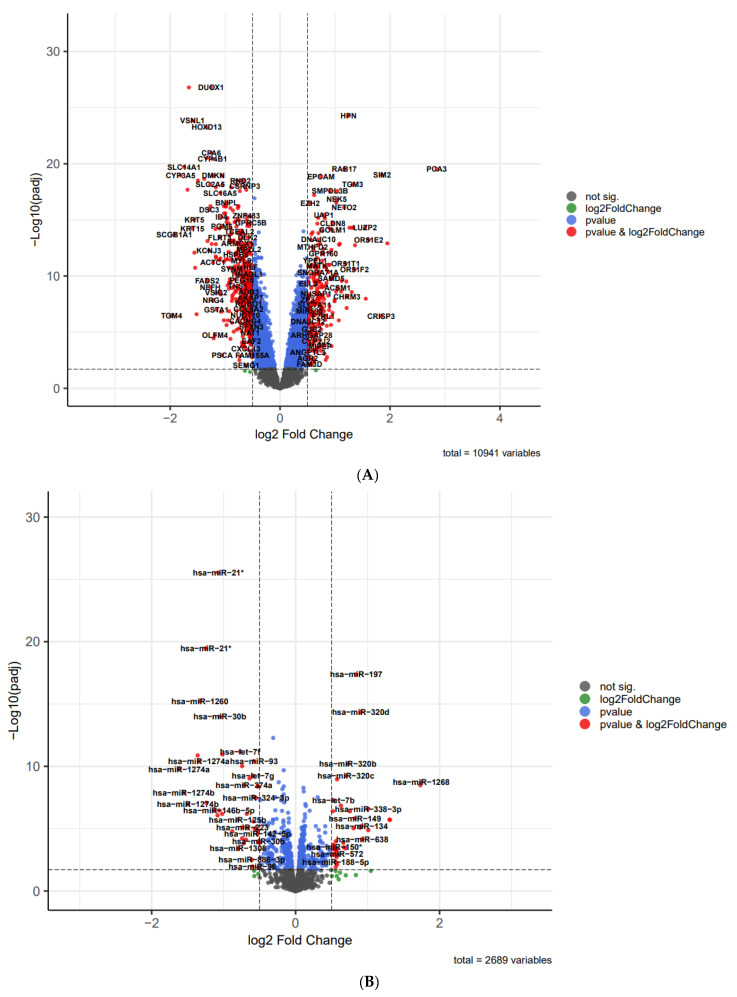
The volcano plot of DElncRNAs, mRNAs DEmRNAs (**A**) and DEmiRNAs (**B**); horizontal axis, log_2_(FC); vertical axis, −log_10_(adjusted *p* value).

**Figure 5 cells-11-03776-f005:**
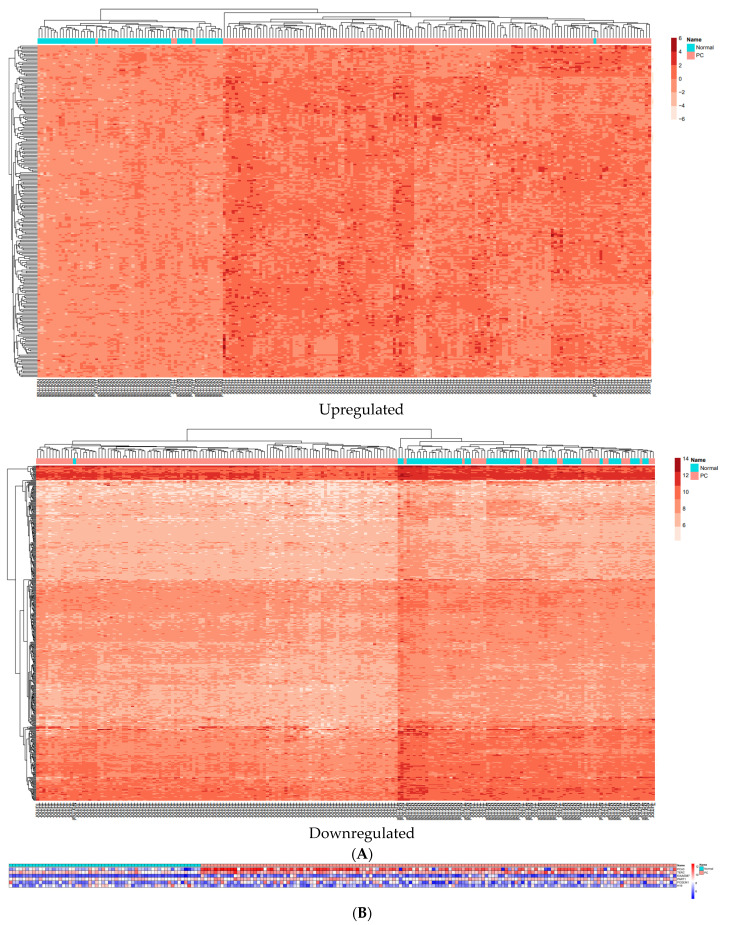
(**A**) Two heatmaps of DEmRNAs between prostate samples and normal samples; horizontal axis, the samples; vertical axis, DEmRNAs. (**B**) The heatmap depicts the expression of DElncRNAs.

**Figure 6 cells-11-03776-f006:**
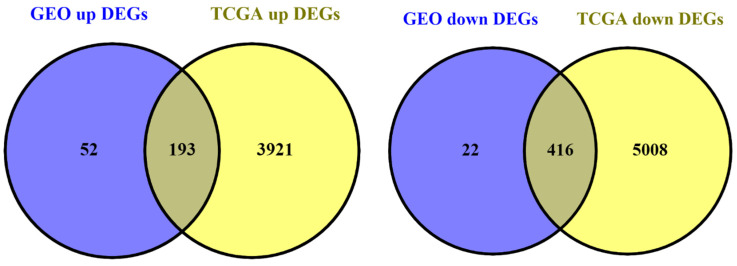
Venn diagram. The number of hub genes in PPI and ceRNA networks and TCGA-PRAD DEGs.

**Figure 7 cells-11-03776-f007:**
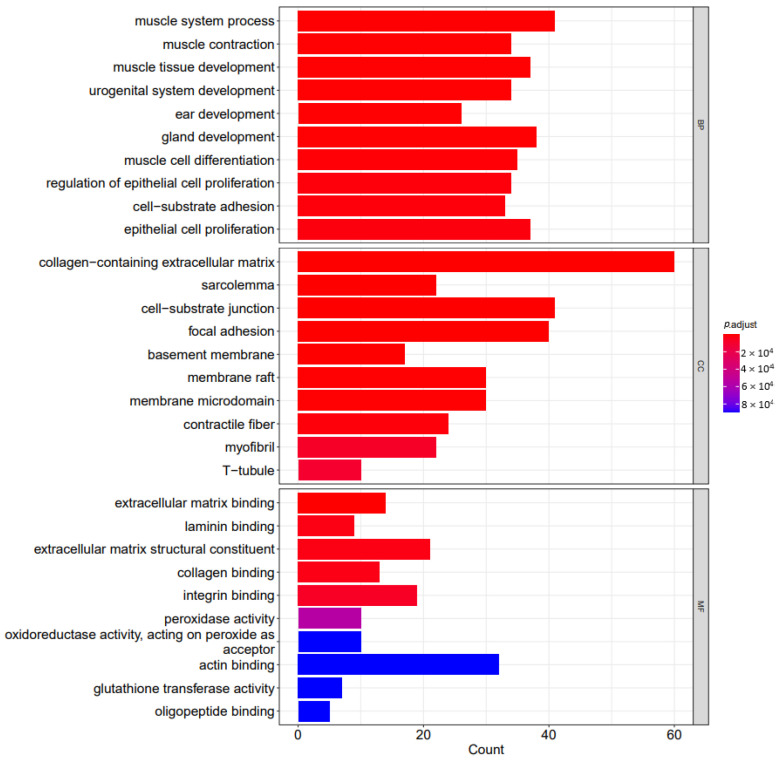
The bar plots of the top 10 enriched functions include BP (biological process), CC (cellular component), and MF (molecular function). The gene-set count is shown on the *X* axis, while the gene-set function is shown on the *Y* axis; the bar color, which ranges from red (most significant) to blue (least significant), indicates the adjusted *p* value.

**Figure 8 cells-11-03776-f008:**
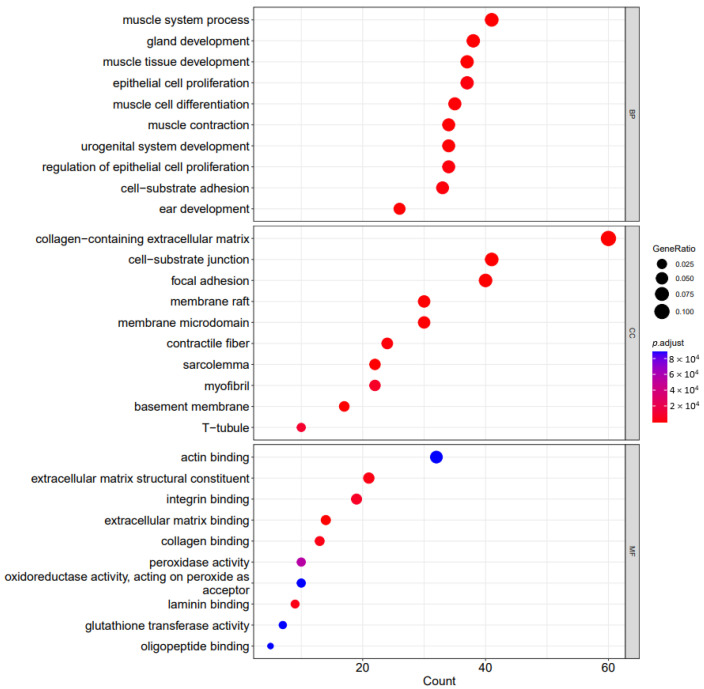
The dot plots of top 10 enriched functions. The gene-set count is shown on the *X* axis, while the gene-set function is shown on the *Y* axis. The color of the dot reflects the adjusted *p* value, and ranges from dark blue (most significant) to red (least significant). GeneRatio is represented by the dot size.

**Figure 9 cells-11-03776-f009:**
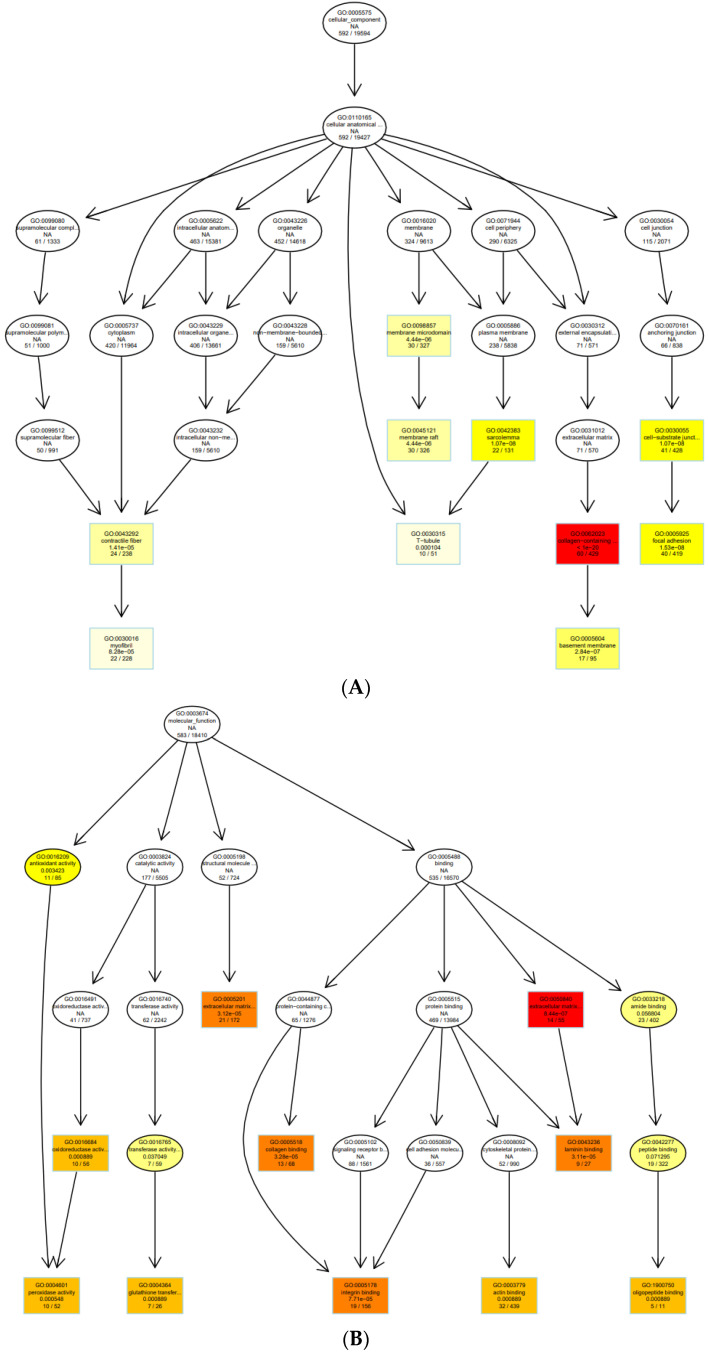
GO graph visualization of the top GO terms enriched. (**A**) A GO sub-graph has been created using the top 30 GO keywords in the category “Cellular Component”. (**B**) A GO sub-graph has been created using the top 30 GO keywords in the category “Molecular Function”. (**C**) A GO sub-graph has been created using the top 30 GO keywords in the category “Biological Process”. Boxes indicate the most significant terms. From dark red (most significant) to light yellow (least significant), the color of the box indicates the relative significance.

**Figure 10 cells-11-03776-f010:**
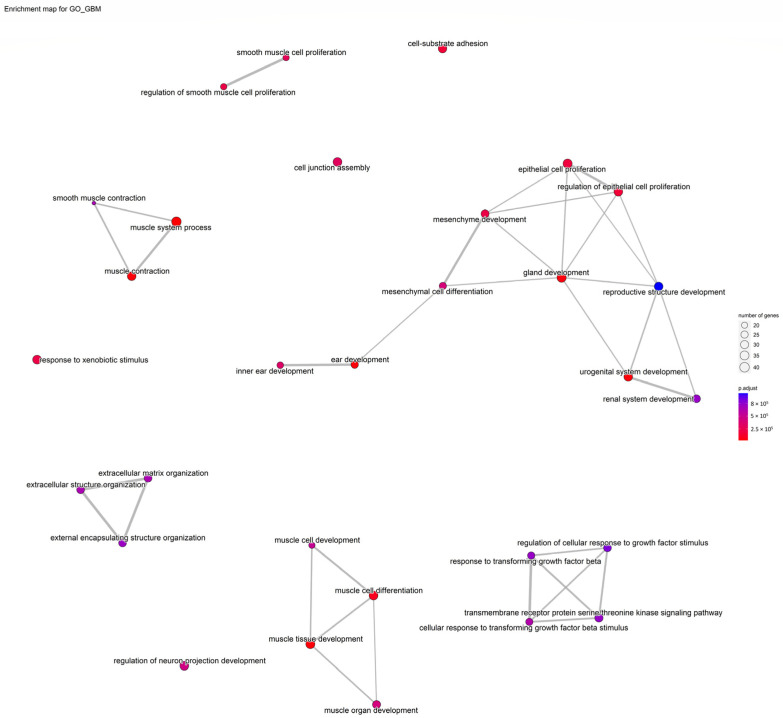
A network of GO terms.

**Figure 11 cells-11-03776-f011:**
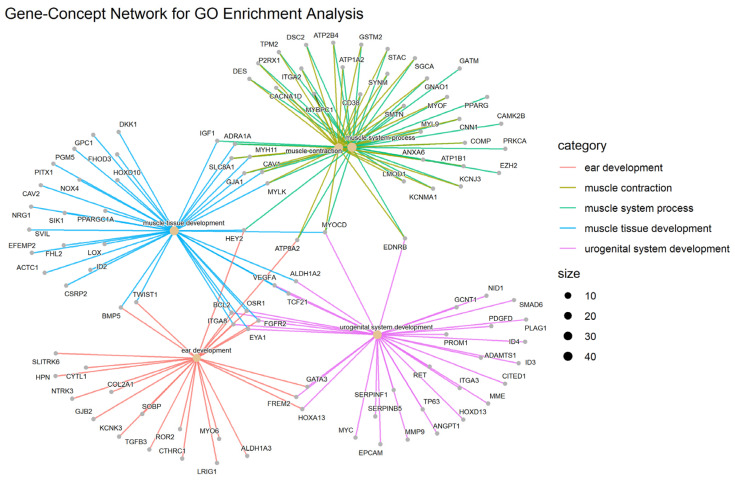
Top 5 GO terms as a network plot. These GO terms are connected to genes in this graph. The connection of genes to the corresponding GO is marked with a special color; there are more genes for a specific GO term if the dot relating to it is bigger.

**Figure 12 cells-11-03776-f012:**
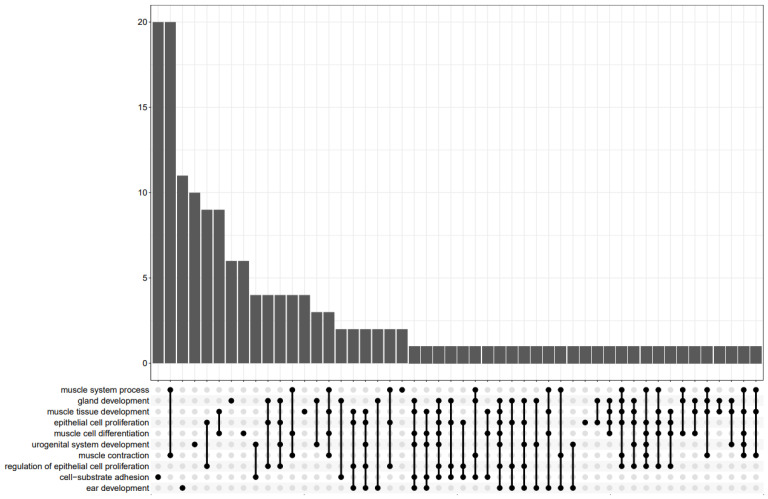
Upset plot of 10 GO terms.

**Figure 13 cells-11-03776-f013:**
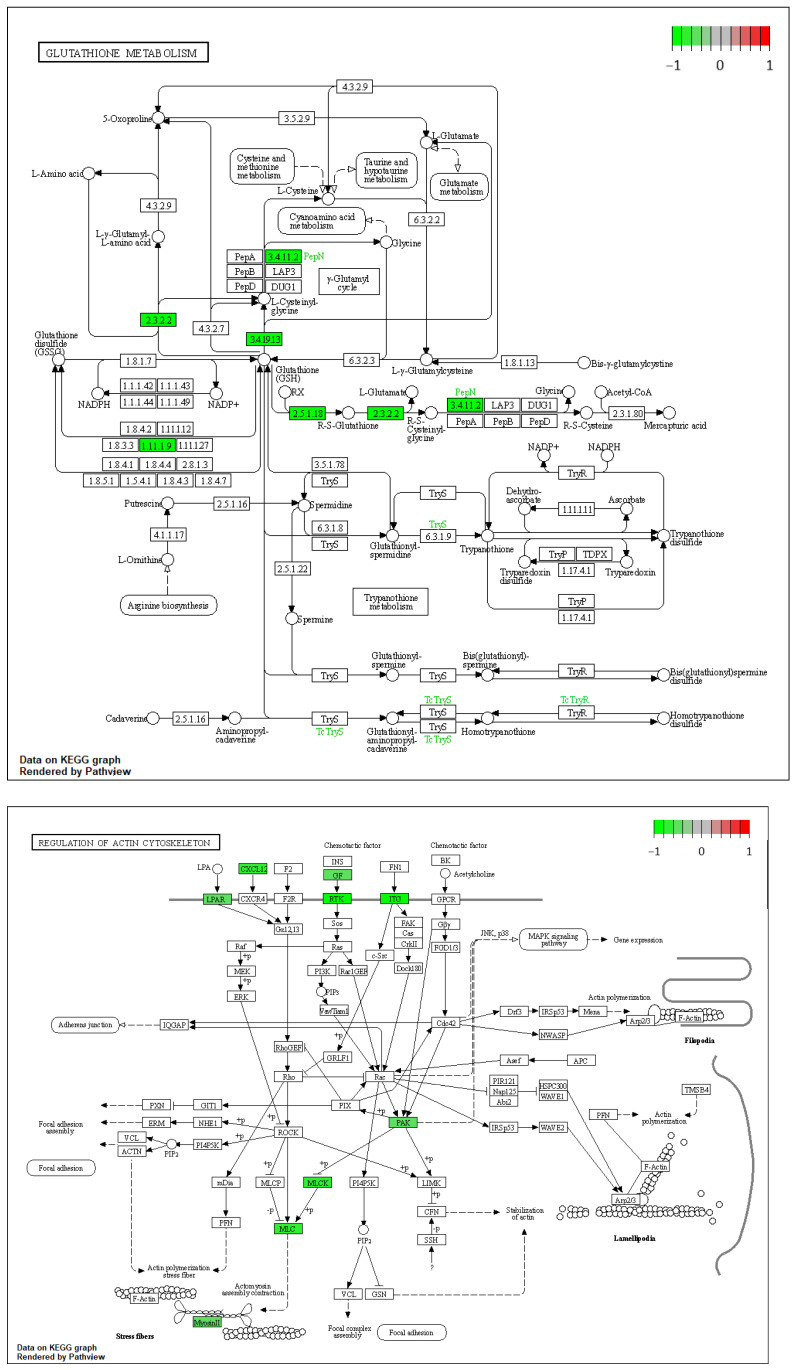
Visualization of the pathways [25]. Green boxes are downregulated genes.

**Figure 14 cells-11-03776-f014:**
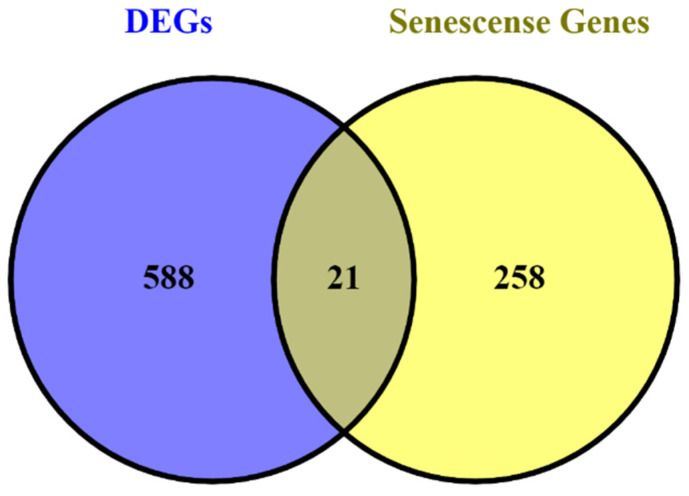
A total of 21 DEGs are related to cell senescence.

**Figure 15 cells-11-03776-f015:**
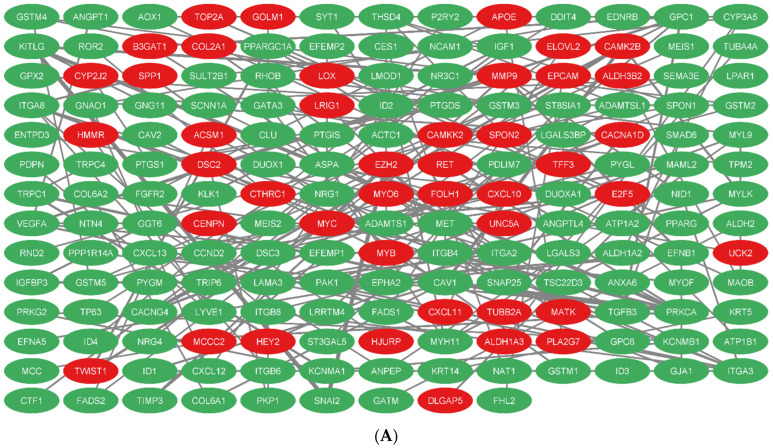
PPI network of DEmRNAs. (**A**) Total PPI network, (**B**) subnetwork of hub genes.

**Figure 16 cells-11-03776-f016:**
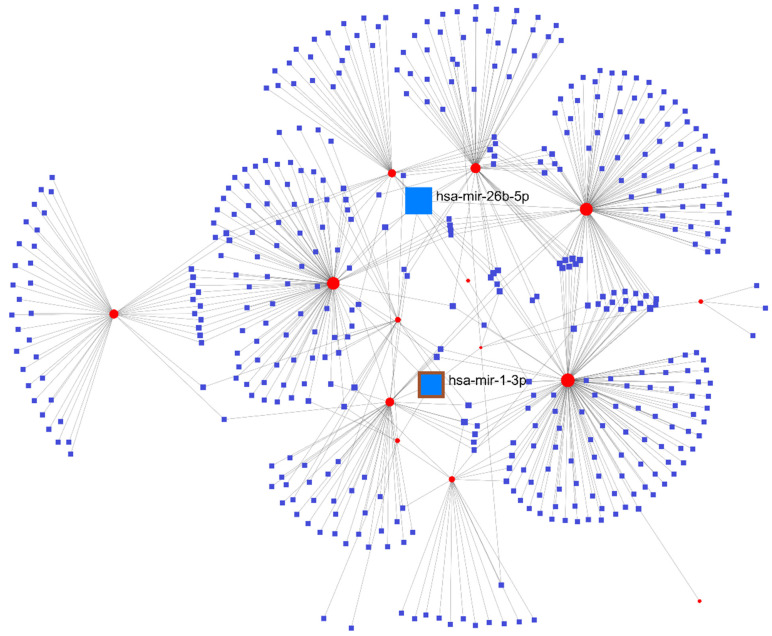
The network of miRNA-hub genes. Circles represent the hub gene, while the squares represent miRNAs targeting the hub genes.

**Figure 17 cells-11-03776-f017:**
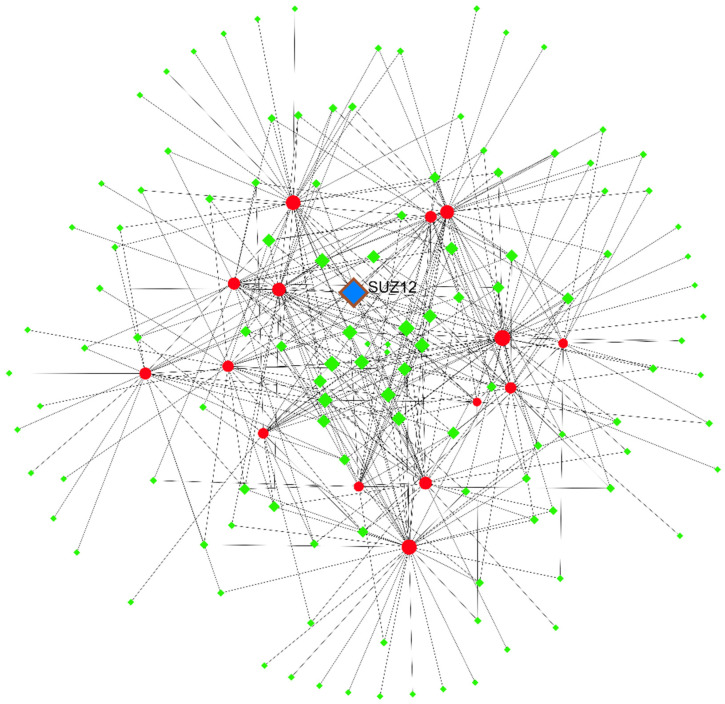
The network of TF-hub genes. Circles show the hub gene, while the diamonds represent TFs targeting the hub genes.

**Figure 18 cells-11-03776-f018:**
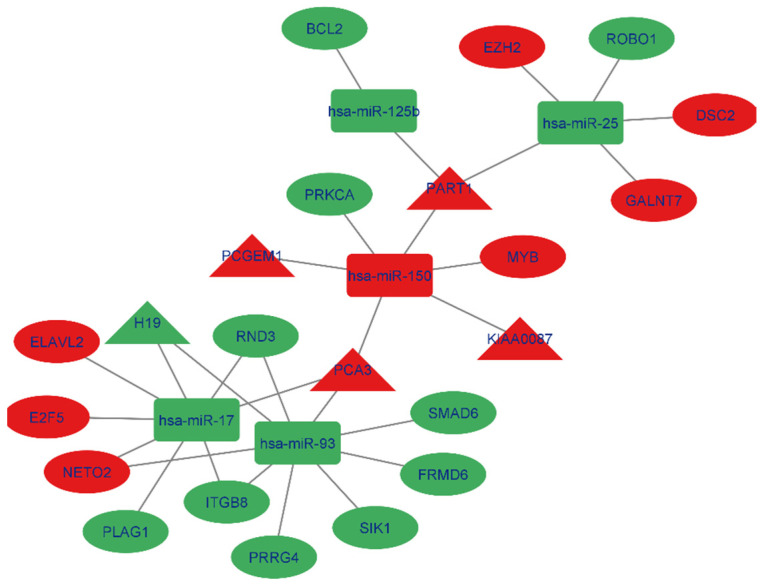
Red nodes signify a strong expression level, while green nodes signify a low level of expression. Ellipses signify protein-coding genes; rectangles signify miRNAs; triangles represent lncRNAs; gray edges specify lncRNA–miRNA–mRNA interactions.

**Figure 19 cells-11-03776-f019:**
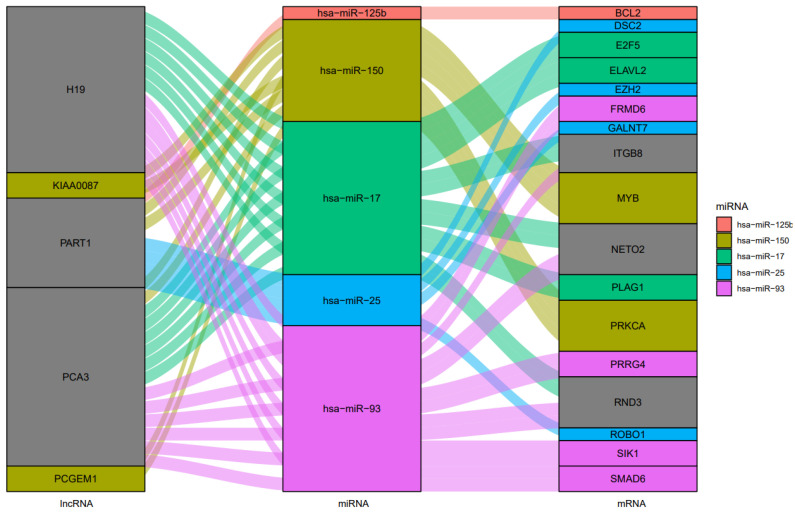
The ceRNA network in prostate cancer is shown by a Sankey diagram. Each rectangle represents a gene, and depending on the size of the rectangle, the degree of relationship between each gene is shown.

**Figure 20 cells-11-03776-f020:**
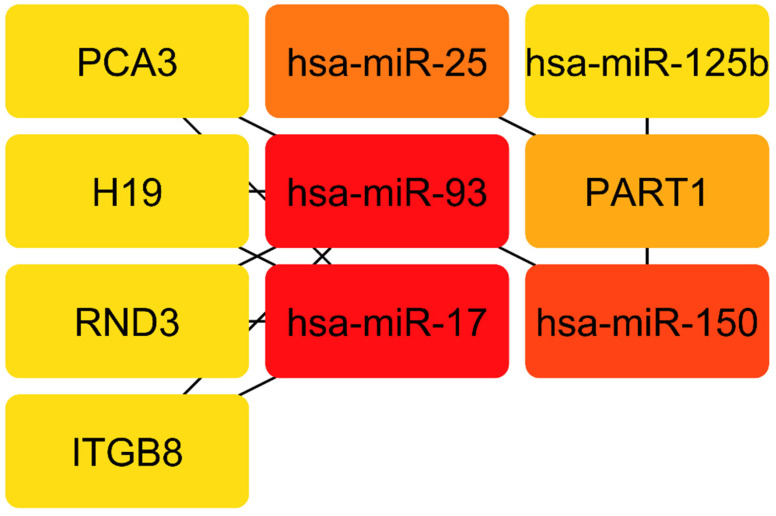
Top 10 genes with highest closeness centrality in ceRNA network.

**Figure 21 cells-11-03776-f021:**
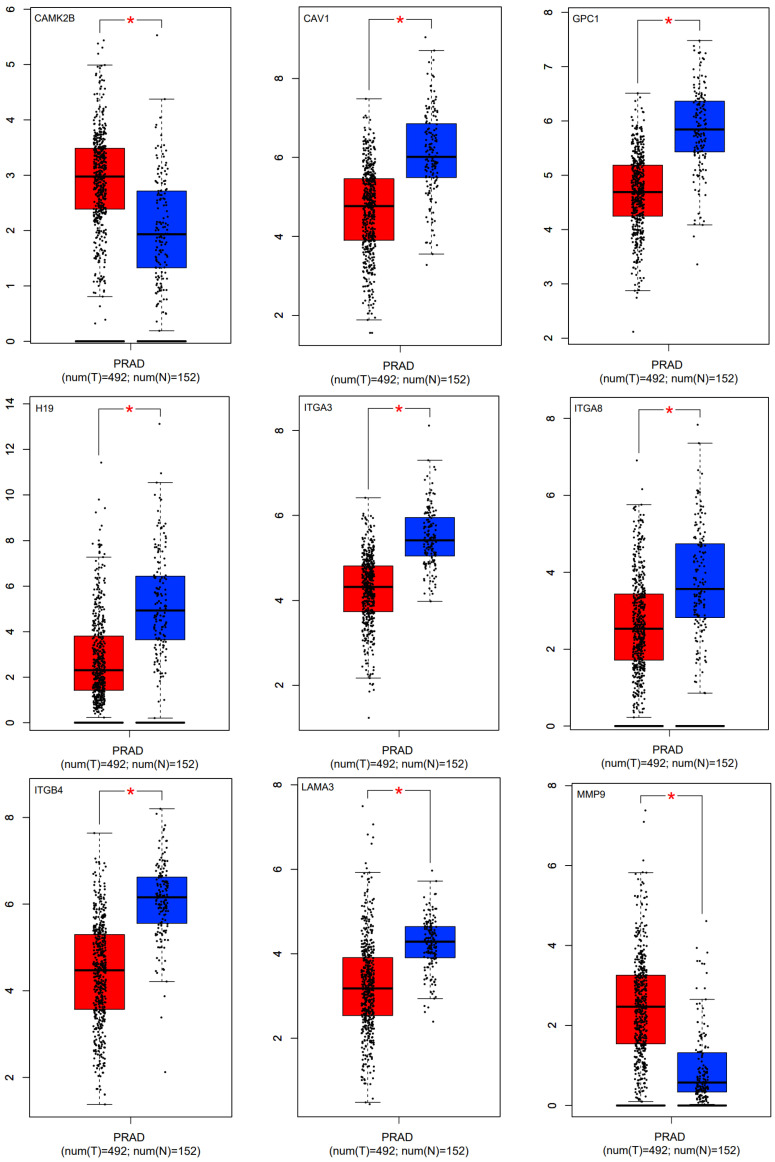
Box plots of the gene expression of hub genes in prostate and normal samples based on TCGA. Red and blue boxes show the gene expression of hub genes in prostate and normal samples, respectively. The red star in some boxplots means that the expression difference between the samples is significant.

**Figure 22 cells-11-03776-f022:**
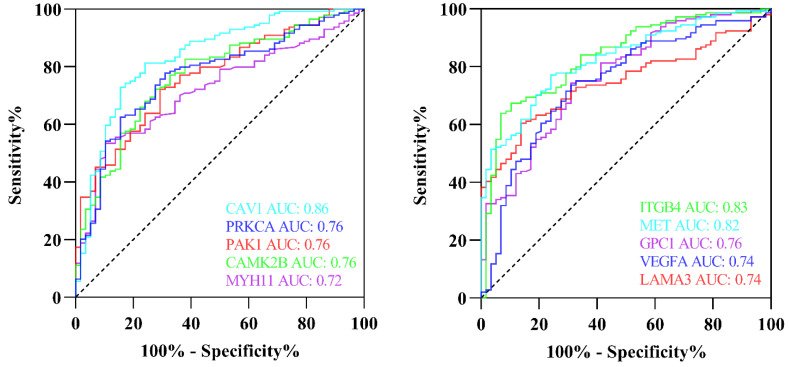
Appraisal of the hub genes. ROC curves for 22 hub genes. The calculated AUC area under the curve for dataset ranged from 0.7 to 1.

**Table 1 cells-11-03776-t001:** Information of datasets.

Datasets.	Platform	Use	Patient	Control	Tissue
GSE88808	GPL22571	DEmRNA-DElncRNA	49	49	Prostate
GSE200879	GPL32170	DEmRNA-DElncRNA	128	9	Prostate
GSE60117	GPL13264	DEmiRNA	56	21	Prostate

**Table 2 cells-11-03776-t002:** The top 10 up- and down-regulated DEmRNAs between prostate and normal samples.

Down-Regulated			Up-Regulated		
DEmRNA	Log FC	Adjusted *p* Value	DEmRNA	Log FC	Adjusted *p* Value
TGM4	−1.81445890301	1.88134991221065 × 10^−6^	TDRD1	1.96438775509	2.94051508709133 × 10^−13^
SCGB1A1	−1.79767675328	1.1360308699886 × 10^−12^	CRISP3	1.87368847071	3.24726996164822 × 10^−7^
CYP3A5	−1.65579619243	3.31725286897387 × 10^−17^	SIM2	1.74655085898	4.647251888494 × 10^−19^
SLC14A1	−1.65191691277	4.64209378540325 × 10^−18^	OR51E2	1.70867540991	6.13073882233394 × 10^−15^
UPK1A	−1.63034519258	7.18623407849526 × 10^−17^	ERG	1.61588303371	7.65718475641531 × 10^−9^
DAPL1	−1.53220612346	1.9314597808472 × 10^−11^	LUZP2	1.56541422855	2.35873230610011 × 10^−16^
DUOXA1	−1.51899377232	2.05445836986054 × 10^−26^	OR51F2	1.42414934014	7.53873927742971 × 10^−12^
VSNL1	−1.47785936294	1.15093949751989 × 10^−20^	SERPINA3	1.36101547484	5.22437085901616 × 10^−10^
CD177	−1.45668628491	6.69536496429305 × 10^−11^	SLC45A2	1.32890185136	6.82938830563377 × 10^−13^
KRT15	−1.45323249293	8.37913282536119 × 10^−13^	DLX1	1.28252404614	3.49922286189955 × 10^−14^

**Table 3 cells-11-03776-t003:** The up- and down-regulated DElncRNAs between prostate and normal samples.

Down-Regulated			Up-Regulated		
DElncRNA	Log FC	Adjusted *p* Value	DElncRNA	Log FC	Adjusted *p* Value
H19	−1.0381434825	4.13318956179463 × 10^−8^	PCA3	3.0652524078	2.06486869720981 × 10^−25^
			PCGEM1	0.6952150040	0.0165229458698012
			KIAA0087	0.6870034897	8.50318057764249 × 10^−6^
			TERC	0.6575339188	6.93708899641187 × 10^−7^
			PART1	0.5929794137	1.36877756290811 × 10^−5^

**Table 4 cells-11-03776-t004:** The top 10 up- and downregulated DEmiRNAs between prostate and normal samples.

Down-Regulated			Up-Regulated		
DEmiRNA	Log FC	Adjusted *p* Value	DEmiRNA	Log FC	Adjusted *p* Value
hsa-miR-1274a	−1.6191209	1.72 × 10^−10^	hsa-miR-1268	1.7455355	1.87 × 10^−9^
hsa-miR-1274b	−1.5536019	1.38 × 10^−8^	hsa-miR-1207-5p	1.3106131	1.96 × 10^−6^
hsa-miR-1260	−1.3607157	1.32 × 10^−11^	hsa-miR-205	1.0448365	0.0245
hsa-miR-21	−1.2456683	3.34 × 10^−20^	hsa-miR-338-3p	1.0091872	2.56 × 10^−7^
hsa-miR-1308	−1.2394549	8.32 × 10^−8^	hsa-miR-638	1.0089855	1.33 × 10^−5^
hsa-miR-142-3p	−1.1264728	3.62 × 10^−7^	hsa-miR-134	0.917789	6.14 × 10^−6^
hsa-miR-720	−1.0837599	8.61 × 10^−7^	hsa-miR-320d	0.905801	4.53 × 10^−15^
hsa-miR-146b-5p	−1.0641932	3.47 × 10^−7^	hsa-miR-197	0.8431693	4.21 × 10^−18^
hsa-miR-30b	−1.0514609	1.03 × 10^−14^	hsa-miR-149	0.8239717	1.47 × 10^−6^
hsa-miR-150	−0.8801092	1.64 × 10^−5^	hsa-miR-214	0.7540948	4.05 × 10^−7^

**Table 5 cells-11-03776-t005:** Down-regulated and up-regulated pathways.

Down-Regulated		Up-Regulated	
Pathway	*p* Value	Pathway	*p* Value
Glutathione metabolism	0.009743969		
Regulation of actin cytoskeleton	0.041300676		

**Table 6 cells-11-03776-t006:** The information of hub genes in PPI network.

Hub Gene	Adjusted *p* Value	Log2FC	Degree	Closseness Centrality	Betweenness Centrality
ITGA2	6.0533584885592 × 10^−8^	−0.5208186242	10	0.23939393939393938	0.09017016738535721
CAV1	8.91282635281604 × 10^−13^	−0.8409077603	9	0.29924242424242425	0.4624758504505339
ITGA3	7.60154283187703 × 10^−10^	−0.5037277367	9	0.24687499999999998	0.06612958068654272
PRKCA	1.16610481754651 × 10^−8^	−0.5112308295	7	0.3038461538461538	0.407488292298419
ITGB4	7.82620045760532 × 10^−13^	−0.7272225278	6	0.2743055555555556	0.16944869476515048
VEGFA	1.39662632691052 × 10^−5^	−0.6752232048	6	0.2532051282051282	0.1266603298248868
ITGA8	8.22863554284887 × 10^−6^	−0.5563216624	6	0.23099415204678364	0.057575617069287974
MET	2.90476617068553 × 10^−11^	−0.6406154443	6	0.2438271604938272	0.03557286595261279
MMP9	0.00114089353601013	0.5468156204	5	0.26158940397350994	0.16307398649170796
MYH11	9.37807316500015 × 10^−5^	−0.5708966261	5	0.20954907161803715	0.037650113599480686
GPC1	8.55395903416247 × 10^−9^	−0.5393742260	5	0.2438271604938272	0.11728875906091096
CAMK2B	1.18776167986803 × 10^−7^	0.6565135653	5	0.2507936507936508	0.2030725954776588
LAMA3	6.50066323396685 × 10^−5^	−0.5027987850	5	0.24687499999999998	0.05840017928625522
PAK1	4.26292437544967 × 10^−8^	−0.5319966868	5	0.24085365853658536	0.09575277043631478
ITGB6	5.08198308078253 × 10^−13^	−1.0073168769	4	0.20256410256410257	0.0

**Table 7 cells-11-03776-t007:** The MiRcode database showed interactions between 5 DElncRNAs and 5 DEmiRNAs.

lncRNA	miRNA
PCA3, PART1, KIAA0087, PCGEM1	hsa-miR-150
PCA3	hsa-miR-425
PCA3, KIAA0087, PART1	hsa-miR-199a-5p
PCA3, H19	hsa-miR-17
PCA3, PART1	hsa-miR-143
PART1	hsa-miR-25
PART1	hsa-miR-125b
H19, PCA3	hsa-miR-93
PCA3, KIAA0087	hsa-miR-96
PCA3, KIAA0087, PART1	hsa-miR-214
H19. PCA3	hsa-miR-22
KIAA0087, PART1	hsa-miR-30b
KIAA0087, PCGEM1	hsa-miR-375
KIAA0087, H19	hsa-miR-338-3p
KIAA0087, PART1	hsa-miR-10a
KIAA0087, PART1	hsa-miR-133b
PART1	hsa-miR-25
PART1	hsa-miR-142-3p
PART1	hsa-miR-30c
PART1, PCGEM1	hsa-miR-205

**Table 8 cells-11-03776-t008:** miRWalk (miRTarBase, TargetScan and miRDB filters) database showed interactions between 5 DEmiRNAs and 17 DEmRNAs.

miRNA	mRNA
hsa-miR-150	PRKCA, MYB
hsa-miR-17	ITGB8, E2F5, ELAVL2, PLAG1, RND3, NETO2
hsa-miR-25	DSC2, ROBO1, GALNT7, EZH2
hsa-miR-125b	BCL2
hsa-miR-93	ITGB8, PRRG4, RND3, FRMD6, SIK1, NETO2, SMAD6

## Data Availability

The analyzed data sets generated during the study are available from the corresponding author on reasonable request.

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
