# Peer review of "LncRNA/miRNA/mRNA Network Introduces Novel Biomarkers in Prostate Cancer"

_cells, 2022, doi:10.3390/cells11233776_

Round 1

Reviewer 1 Report

The authors presented a very interesting bioinformatics study that uses many modern methods in silico. The work was carried out at a high level and can be used as a model for similar studies in other diseases or as a starting point for the selection of genes for their detailed study and confirmation on biological samples of prostate cancer.

The comments relate to the figures in the manuscript and minor revisions of the English language, which the authors can make themselves.

1. In some cases, "hsa" in microRNA names has become "has". For example, lines 25, 26, 320, 343, 397.

2. Also check the spelling of some words in lines 243 and 259.

3. In line 288, the authors' notes related to the design of the figure remained.

4. Figure 18. Some mRNA and lncRNA names are not readable. Change the font size or color, or shift the text. The difference in the thickness or color of the lines denoting betweenness is not visible.

Author Response

The authors presented a very interesting bioinformatics study that uses many modern methods in silico. The work was carried out at a high level and can be used as a model for similar studies in other diseases or as a starting point for the selection of genes for their detailed study and confirmation on biological samples of prostate cancer.

The comments relate to the figures in the manuscript and minor revisions of the English language, which the authors can make themselves.

  1. In some cases, "hsa" in microRNA names has become "has". For example, lines 25, 26, 320, 343, 397.

Response: Thanks for your comment. We corrected the names of the miRNAs.

  1. Also check the spelling of some words in lines 243 and 259.

Response: Thanks for your comment. We checked them and made the necessary changes.

  1. In line 288, the authors' notes related to the design of the figure remained.

Response: Thanks for your comment. We edited this part.

  1. Figure 18. Some mRNA and lncRNA names are not readable. Change the font size or color, or shift the text. The difference in the thickness or color of the lines denoting betweenness is not visible.

Response: Thanks for your suggestion. We changed the font size and color of the labels. Also, we removed the description about the thickness of the edges.

Reviewer 2 Report

The authors have reported about the identification of an array of biomamarkers (ceRNA) to improve the identification of PCa.

The experimental designs and results are well explained. I suggest to the authors to better characterize three main points. A) in introduction: the limitations of PSA which is not quoted in this paper as recommended biomarker for PCa. You can find relevant suggestions in this paper reporting also about the recent updates of current Guidelines concerning PSA-based diagnostic of PCA. Ferraro S, et al. Serum prostate specific antigen (PSA) testing for early detection of prostate cancer: Managing the gap between clinical and laboratory practice.  Clin Chem 2021;67:602-9.

B)In introduction/discussion:The evidence that high grade PCa is likely on genetic basis and this is relevant for your research, whereas low grade disease is more likely associated to environmental factors. These papers may be of aid:

Gleason grade progression is uncommon. Penney KL, et al.Cancer Res. 2013 Aug 15;73(16):5163-8.

Roudier MP, et al. Phenotypic heterogeneity of end-stage prostate carcinoma metastatic to bone. Hum Pathol. 2003 Jul;34(7):646-53. 

C)For reasons of cost-effectiveness, this pannel should be applied to individuals at risk of high grade PCa, likely based on PSA levels. According to current guidelines recommendations, more recently a PSA-based algorithm for predicting high grade tumors has been developed. 

Ferraro S, et al.Definition of Outcome-Based Prostate-Specific Antigen (PSA) Thresholds for Advanced Prostate Cancer Risk Prediction. Cancers (Basel) 2021;13:3381.

You could think to apply the ceRNA as second level test to patients with PSA levels>6 µg/L.

Author Response

The authors have reported about the identification of an array of biomamarkers (ceRNA) to improve the identification of PCa.

The experimental designs and results are well explained. I suggest to the authors to better characterize three main points. A) in introduction: the limitations of PSA which is not quoted in this paper as recommended biomarker for PCa. You can find relevant suggestions in this paper reporting also about the recent updates of current Guidelines concerning PSA-based diagnostic of PCA. Ferraro S, et al. Serum prostate specific antigen (PSA) testing for early detection of prostate cancer: Managing the gap between clinical and laboratory practice. Clin Chem 2021;67:602-9.

Response: We explained this point and cited the mentioned paper.

  1. B) In introduction/discussion: The evidence that high grade PCa is likely on genetic basis and this is relevant for your research, whereas low grade disease is more likely associated to environmental factors. These papers may be of aid:

Gleason grade progression is uncommon. Penney KL, et al.Cancer Res. 2013 Aug 15;73(16):5163-8.

Roudier MP, et al. Phenotypic heterogeneity of end-stage prostate carcinoma metastatic to bone. Hum Pathol. 2003 Jul;34(7):646-53.

Response: We cited these papers and explained the mentioned notes.

  1. C) For reasons of cost-effectiveness, this pannel should be applied to individuals at risk of high grade PCa, likely based on PSA levels. According to current guidelines recommendations, more recently a PSA-based algorithm for predicting high grade tumors has been developed.

Ferraro S, et al. Definition of Outcome-Based Prostate-Specific Antigen (PSA) Thresholds for Advanced Prostate Cancer Risk Prediction. Cancers (Basel) 2021;13:3381.

Response: We cited this paper and explained the note.

You could think to apply the ceRNA as second level test to patients with PSA levels>6 µg/L.

Response: We suggested this note in the discussion.